# Investigation of the Zero-Frequency Component of Nonlinear Lamb Waves in a Symmetrical Undulated Plate

**DOI:** 10.3390/s24154878

**Published:** 2024-07-27

**Authors:** Xiaoqiang Sun, Guoshuang Shui

**Affiliations:** 1College of General Education, Chongqing Industry Polytechnic College, Chongqing 401120, China; sunxq@cqipc.edu.cn; 2Department of Mechanics, Beijing Jiaotong University, Beijing 100044, China

**Keywords:** zero-frequency component, structure health monitoring, undulated plate, early-stage damages

## Abstract

When an ultrasonic pulse propagates in a thin plate, nonlinear Lamb waves with higher harmonics and a zero-frequency component (ZFC) will be generated because of the nonlinearity of materials. The ZFC, also known as the static displacement or static component, has its unique application on the evaluation of early-stage damages in the elastic symmetrical undulated plate. In this study, analysis of the excitation mechanism of the ZFC and the second harmonic component (SHC) was theoretically and numerically investigated, and the material early-stage damage of a symmetrical undulated was characterized by studying the propagation of nonlinear Lamb waves. Both the ZFC and SHC can be effectively employed in monitoring the material damages of the undulated plate in its early stage. However, several factors must be considered for the propagation of the SHC in an undulated plate because of the geometric curvature and interference between the second harmonics during propagation, preventing efficient application of this technique. If the fundamental wave can propagate in the plate regardless of the plate boundary conditions, an accumulative effect always exists for the ZFC in a thin plate, indicating that the ZFC is independent of the structural geometry. This study reveals that the ZFC-based inspection technique is more efficient and powerful in characterizing the damages of a symmetrical undulated plate in the early stage of service compared to the second harmonic method.

## 1. Introduction

Undulated plates or shell structures have unique physical characteristics and are widely employed in several engineering applications, such as plate heat exchangers that increase acceptance in a two-phase flow [1]. When using these structures, they can be subjected to high stresses, temperatures, and pressures, which cause fatigue, plasticity, corrosion, and creep damage. Consequently, when plate performance degrades, it results in explosions, fractures, and potential leakage of dangerous substances. With continuous development, several relatively mature nondestructive testing (NDT) and evaluation technologies are available, including liquid penetration, magnetic powder, eddy current, X-ray, and linear ultrasound techniques [2]. However, these traditional techniques can only detect macroscopic defects and damage in materials in the order of millimeters, such as macroscopic cracks, impurities, delamination, and pores. For early-stage damage at a micro- and nanoscale, such as material dislocation and slip, they are not sufficiently sensitive to conventional linear ultrasonic waves. Previous studies have shown that early-stage damage accounts for most of the material’s service life before macroscopic damage occurs in a structure. When fatigue damage occurs in a material, 80–90% of the material’s entire fatigue life is due to the performance degradation of early-stage damage [3,4]. Consequently, research, development, and optimization of NDT methods for early-stage damage in materials are important for ensuring safety in engineering applications.

Extensive research has revealed that nonlinear ultrasound detection technology is highly sensitive to microstructural changes in materials, such as the dislocation and microcrack density, and can effectively detect and evaluate early-stage material damage [5]. Nonlinear Lamb wave–based detection technology developed for plate- and shell-like structures has gained increasing attention due to its numerous advantages, such as a long propagation distance, a long detection range, and a sensitivity to early-stage damage. Nonlinear Lamb wave–based techniques includes the second harmonic (or higher harmonic) method [6,7,8,9,10,11,12,13,14,15,16,17,18,19,20,21,22,23,24,25,26,27,28,29,30,31], the sum- and difference-frequency (mixing wave) method [32,33,34,35,36,37,38,39,40,41,42,43,44,45,46,47], and the zero-frequency component (ZFC) method [48,49,50,51,52,53,54,55,56,57,58,59,60,61,62,63,64,65,66].

### 1.1. Second Harmonic Method

Several studies employing nonlinear Lamb waves have been published in the field of early-stage damage NDT using the second harmonic method. However, methods using the second harmonic component (SHC) have limitations. Particularly, phase velocity matching [7,8] causes difficulty in the effective application of the SHC-based method. If this matching condition is not satisfied, rather than a linear accumulative increase with the propagation distance, the SHC behaves similar to a periodic sinusoidal oscillation with the fundamental wave propagation. This is because Lamb waves have dispersive properties, and, except for a few limited choices of frequencies (mode pairs) [24], the phase velocity of the second harmonics is not equal to that of the fundamental waves. During the propagation of fundamental waves, interference between the newly generated second harmonics results in the formation of a periodic variation along the propagation path. For nonlinear Lamb waves, where phase mismatch happens, the first rising part of the second harmonic periodic variation is utilized to measure the nonlinearity of the material. This means that the maximum detection distance is approximately half of the space period of the variation of the second harmonics, significantly limiting the application of the SHC-based method. Thus, more mode pairs and the S0 mode with low frequencies [24] have been proposed when considering the matching of phase velocity to continuously increase the SHC in terms of the fundamental wave propagation distance for greater detection distance.

In Lamb wave–dispersion curves, the lower frequency component can be considered as the fundamental wave and the corresponding higher frequency component can be regarded as the SHC when the higher frequency is twice as much as the lower frequency. As the mode pair selected in advance satisfies the phase matching condition, the energy of the second harmonics linearly accumulates with the propagation distance. Bermes et al. [10,11] utilized the S1–S2 mode pair, with S1 being the first-order symmetrical-mode fundamental wave and S2 being the second-order symmetrical-mode second harmonics, to measure the nonlinearity of materials, including fatigue damage [12], tensile plastic damage [13], temperature fatigue damage [15], and creep damage [18,19,20]. Additionally, more investigations have been conducted on other mode pairs, such as the A2–S4 [21] and S2–S4 [22] mode pairs.

The mode-pair method, which uses Lamb wave dispersion and multimode features, is the foundation for the practical application of the second harmonic–based method. However, these mode pairs are limited, and they cannot meet phase velocity matching over a wide band range. In actual applications, determining the frequency of the excitation signals to strictly match the expected selected frequency can be challenging [24].

To overcome the disadvantages of the dispersion, technical difficulties, and low detection efficiency of mode pairs, Wan et al. [24] proposed a technique employing S0 mode Lamb waves with low frequencies. Because the dispersion of the S0 mode is relatively weak when the frequency is low, it can easily satisfy the condition of phase velocity matching to some extent. Therefore, a longer cumulative propagation distance for the SHC can be investigated. Consequently, an application of the S0 mode in lower frequencies brings more mode pairs when considering the matching conditions, and it exhibits better robustness, weaker dispersion, and easier signal processing. Only the S0 mode signal is considered when it is excited and receives the nonlinear Lamb waves. Considering these points, this enhances the SHC-based detection capabilities and has aroused the interest of researchers in recent years. Low-frequency S0 mode waves could compensate for the limitations of mode pairs when used in microcrack detection [24,25,27]. Moreover, numerous simulations have been conducted for low-frequency S0 mode waves, and the effectiveness of detecting material nonlinearities using the low-frequency S0 mode has been confirmed via experiments and simulations [28]. However, when using the SHC-based method, the frequency range satisfying the matching condition is narrow, and the maximum distance at which the SHC linearly accumulates rapidly decreases as the fundamental frequency increases.

Based on the theoretical calculations and the published investigations [24], the maximum linear accumulation distance for the SHC is approximately 162 mm if the fundamental frequency is 300 kHz. When the fundamental frequency is 500 kHz, the maximum linear cumulative distance of the SHC is limited for actual applications, and the method is not practically feasible. To obtain a detection distance of a half meter, the fundamental frequency must be lower than 200 kHz. At a fixed pulse frequency, the following points should be considered: (1) the lower the fundamental frequency, the longer the propagating length of the pulse wave packet; (2) the actual plate used is not infinitely large, and low-frequency pulses are more likely to cause superimposition of the fundamental component and boundary reflection waves on each other; and (3) for a fixed amplitude, the lower the frequency, the less the energy is contained by the ultrasonic wave, and a significantly low frequency for the fundamental component may not be beneficial for material early-stage damage detection.

### 1.2. Zero-Frequency Component (ZFC)–Based Method

When a wave propagates in a nonlinear medium, in addition to exciting harmonics (sum- and difference-frequency), it also excites the ZFC, which is also known as the static displacement or static component, and has been the focus of several researchers who have noticed this in bulk waves. Stress caused by the ZFC has been discussed in the early 20th century. For example, Brillouin [48] reported that the stress caused by the ZFC was related to changes in the time-averaged momentum of the wave particle, and other researchers [49] stated that the ZFC-induced stress was directly related to the nonlinear properties of materials. In the early 1980s, Cantrell and Yost [52] utilized sensors to collect the ZFC when measuring the nonlinear coefficients of materials, where the ZFC showed as a right-angled triangular. In 2006, Jacob et al. [54] employed lasers to detect the ZFC. Moreover, the ZFC shape excited by a pulse of limited width is approximately similar to that of the fundamental wave pulse packet when the waves are filtered, indicating that the ZFC linearly increases with the energy of the fundamental wave. In 2007, Narasimha et al. [53] used sensors to detect a ZFC-induced strain and proposed a new signal-processing method to extract the ZFC. Subsequently, Narasimha et al., Qu et al., and Jacob et al. [54,55,56,57] conducted excellent research on the ZFC method, laying a solid foundation for effective measurement of the nonlinearity of materials.

However, research on the ZFC is limited compared with investigations on bulk waves. Sun et al. [58] reported a systematic theoretical investigation on the ZFC of a nonlinear Lamb wave for the first time. The study highlighted that the ZFC could be continuously accumulated when non-zero energy flowed from the fundamental component to the ZFC. Rather than the phase-velocity matching being a necessary condition for the accumulation of the SHC, only one condition should be satisfied for the ZFC accumulation of the nonlinear Lamb wave. The integral value of the ZFC excited by the Lamb waves is larger than that of the second harmonics, indicating that the ZFC is symmetric.

In 2018, Wan et al. [59] studied the ZFC excited by Lamb waves via COMSOL simulations, and their study is consistent with the corresponding theoretical results of Sun et al., who further demonstrated that the shape of the ZFC in nonlinear Lamb waves was generally similar to the envelope of the fundamental waves. The study shows that the phase- and group-velocities of the ZFC are equal to those of the fundamental waves, and they are also equal to the velocity value at a zero frequency on the S0-dispersion curve. Subsequently, Sun et al. [60] verified the feasibility of using the ZFC method for the detection of early-stage damages in the plate-like materials via theoretical investigations, simulations, and experiments. Compared with the second harmonic technique, the ZFC-based methodology has several advantages. In 2020, Sun et al. [63] theoretically and numerically investigated the ZFC generated by uniformly distributed random cracks in the solids. It shows that the ZFC in nonlinear buck waves can also be used for microcrack damage detection. Sun et al. [62] verified that lead zirconate titanate (PZT) could be employed to effectively collect the ZFC, exhibiting the possibility of detecting localized plastic damage in the plate material using the ZFC-based experimental method. Furthermore, Deng [64] used conventional transducers to effectively detect the ZFC. Gao et al. [65] conducted experiments to detect the ZFC in the nonlinear Lamb wave, confirming the sustained accumulation characteristics of the ZFC. Chen et al. [66] carried out related studies on the ZFC generated in multilayer structures, and Sun [67] located the mutation damages using the ZFC-based method. Without the phase velocity matching condition, the ZFC in the nonlinear Lamb waves has the characteristics of continuous accumulation, and there are many options for the selection of the frequency of the fundamental waves, substantially extending the potential engineering applications of this ZFC-based technique.

Several important results have been achieved in evaluating the early-stage damage of plate structures using the SHC-, wave mixing–, and ZFC-based detection methods. In recent years, there has been an abundance of publications focusing on zero-frequency applications, which has elevated zero frequency to a prominent and increasingly popular research topic. However, these studies are mainly concerned with the characterization of thin plates with a uniform thickness. Although Hu et al. [68] studied the propagation of the second harmonics in a plate with varying thickness, an effective method for early-stage damage detection of a plate with varying thickness is still not well established. Few investigations have been reported on the early-stage damage detection of plates of nonuniform thickness until now. Consequently, proposing and examining an efficient early-stage damage-detection method for thin plates with nonuniform thicknesses is essential.

This study aims to broaden the application of ZFC- and SHC-based methodologies on the early-stage damage evaluation of symmetrical undulated plates with nonuniform thicknesses. The study provides an in-depth theoretical and numerical investigation of the ZFC of nonlinear Lamb waves, offering new insights into early-stage damage detection, which is crucial for structural safety in engineering applications. The remainder of this paper is organized as follows. Section 2 establishes the theory of the early-stage damage detection of a symmetrical undulated plate based on the ZFC and the second harmonic waves. Section 3 examines the early-stage damage detection of a symmetrical undulated plate based on the ZFC and second harmonic method. The discussion of the results and conclusions are presented in Section 4.

## 2. Theory

Considering the periodically undulated isotropic plate shown in Figure 1, the plate has periodic geometric undulations, which can be described as $x_{3}(U)=h+A\sin((2\pi /\Lambda )x_{1})$ and $x_{3}(L)=-A\sin((2\pi /\Lambda )x_{1})-h$, where 2*h* is the average thickness of the undulated plate, *A* is the amplitude of sine function for the periodic surface (A<h), Λ is the spatial length of the periodicity for the outer surface, and $L_{n}$ is the variation period of the second harmonics along the propagating distance for a rectangular cross section with thickness 2*h*.

For the Lamb waves propagating with finite amplitude in the isotropic and homogeneous thin plate of uniform thickness and stress-free surfaces, the ZFC solution can be expressed as [67]
(1)$$A^{0}(x_{1})=cx_{1},\;c\equiv \frac{f_{n}}{4P_{mn}},$$
where $A^{0}(x_{1})$ is the ZFC magnitude and *c* is a constant parameter defined by $f_{n}/4P_{mn}$, which is called the power flux ratio. We can relate this ratio to elastic constants of the third order and plate thickness. Detailed expressions of *f_n_* and *P_mn_* are given in [67]. Evidently, the ZFC magnitude increases with the propagating distance along the $x_{1}$- direction linearly.

For the undulate plate, the energy of the fundamental waves is assumed to not decay; thus, the amplitude remains unchanged during the propagation. Considering the S0 mode Lamb wave for simplicity, the power flow term ($f_{n}$) in Equation (1) can be related to the undulated plate thickness $h(x_{1})=x_{3}(U)-x_{3}(L)$. If $f_{0}$ is considered as the power flow of the ZFC in a uniform plate with thickness 2*h*, ignoring the energy attenuation of the primary wave, we obtain
(2)$$f_{n}(x_{1})=\frac{f_{0}\cdot 2h}{h(x_{1})}=\frac{f_{0}\cdot h}{h+A\sin\left(2\pi x_{1}/\Lambda \right)}.$$

By substituting Equation (2) into Equation (1), the ZFC magnitude can be expressed as
(3)$$A^{0}(x_{1})=\int_{0}^{x_{1}}[ch/(h+A\sin((2\pi /\Lambda )x_{1})]dx_{1}=\int_{0}^{x_{1}}I(x_{1})dx_{1},$$
where
(4)$$I(x_{1})\equiv ch/(h+A\sin((2\pi /\Lambda )x_{1}).$$

Because of the periodic undulation of the plate, the power-related term is no longer constant, and the integrand $I(x_{1})$ in Equation (3) exhibits high oscillation. As shown in Figure 2, $I(x_{1})$ is plotted as a repetitive oscillation function of the propagating distance $x_{1}$ when c = 1, *h* = 1 mm, *A* = 0.6 mm, and Λ = 40 mm. The integration of $I(x_{1})$ with respect to $x_{1}$ (magnitude of the ZFC) is presented in Figure 3. The ZFC magnitude, $A^{0}(x_{1})$, increases with the wave propagating distance $x_{1}$. Despite the nonuniform thickness, the cumulative growth of the ZFC magnitude is hardly affected by periodic undulation of the plate, and the undulation of the plate does not change the cumulative characteristics of the ZFC. Although the ZFC and fundamental waves have different phase velocities because of the dispersion, the phase and group velocities of the ZFC are equal and the phases of the ZFCs are the same. Consequently, the newly generated ZFCs and previously generated components will interfere with each other. Moreover, the enhanced interference is always sustained as long as the energy of the fundamental wave propagates through the plate, regardless of the plate geometry.

The evaluation of the material early-stage damage in a symmetrical undulated plate based on the ZFC method is not difficult because the detection procedure shows no special difference between a uniform plate and an undulated plate. The following steps were carried out. First, the fundamental wave is excited and loaded onto the undulated plate. Subsequently, the transmitted wave is detected and received at different wave distances along a straight line of wave propagation. Thereafter, the fast Fourier transform is performed on every received signal to obtain the corresponding magnitude of the ZFC, which is finally plotted as a function of the propagation distance and linearly fitted. With this, early-stage damage is evaluated based on the magnitude of the fitting curve slope. Contrary to the conventionally employed amplitude ratio [62,63], the damage factor adopted is only the ZFC magnitude.

However, the solution for the SHC of a nonlinear Lamb wave is difficult for the undulated plate. A structural quasi-phase-matching (SQPM) technique is proposed to enhance the second harmonic intensity in the nonlinear Lamb wave. When the frequency of fundamental waves or the spatial length of this undulated plate changes, the generation and propagation of the second harmonics is comparatively intricate. Consequently, conducting the material early-stage damage evaluation of a symmetrical undulated plate based on the SHC is not easy. The details are provided in Appendix A.

Following SQPM, the second harmonic–based inspection methodology can be developed as follows:Determining the inspection frequency. Assuming the spatial period distance ($L_{n}$) of the second harmonics is equal to the spatial length (Λ) of the undulated plate, the frequency of the fundamental wave can be obtained by solving Equation (A1) in Appendix A.Determining the location of the detection points. To reduce the error, the distance between two adjacent detection points must be $l_{n}$ (or an integral number of $l_{n}$).The fundamental wave in the undulated plate at the selected frequency is excited. Subsequently, the transmitted wave is detected and received at different wave propagation distances along a straight line. The corresponding second harmonic magnitude is obtained by performing the fast Fourier transform on every received signal.

Therefore, the second harmonics magnitude can be presented as a function of the wave propagating distance; thus, early-stage damage can be evaluated based on the magnitude of the fitting curve slope. The damage factor adopted in this study is the magnitude of the second harmonics, not the conventionally employed amplitude ratio [13].

## 3. Numerical Simulation

### 3.1. Simulation Model

By ignoring the damping effect, numerical simulation was carried out using the commercially available ABAQUS software (Version 6.14, Dassault Systems Simulia Corp., Johnston, RI, USA). The hyperelastic constitutive law is considered based on the Landau–Lifshitz model to investigate the undulated plate. The physical parameters of the plate are listed in Table 1, where *ρ* is the mass density; A, B, and C are the third-order elastic constants (TOE) when material nonlinearity is considered; *λ* and *µ* are Lame’s constants; and *E* and *ν* are the Young’s modulus and Poisson’s ratio, respectively. TOE are material properties that describe the nonlinear behavior of an elastic medium.

To increase the efficiency and accuracy for the simulation, the rectangular element of the second order was employed, and the time step and element size are given by [26]
(5)$$\Delta I=\frac{\lambda_{\min}}{20},\;\Delta t=\frac{1}{20f_{\max}},$$
where ΔI is the element size, Δt is the time step, and $\lambda_{\min}$ and $f_{\max}$ are the minimum wavelength and maximum frequency of interest, respectively. In this simulation, the element size and time step are set as 0.2 mm and 5 × 10^−10^ s, respectively.

As shown in Figure 4, the length of the undulated plate (Al-7075-T651) is 2000 mm, where approximately 50 cells are considered. The first cell has a spatial length of Λ=40 mm, and the average thickness of the undulated plate is 2 mm.

Amplitude *A* of the sine function of the plate surface was set as 0.6 mm, with 651 signal detection points evenly distributed at 1 mm interval between $x_{1}=0\;\mathrm{mm}$ and $x_{1}=650\;\mathrm{mm}$. For further discussion, signals were also collected at 11 points on the same cross section of the plate. These points were distributed along the transversal line with a 0.2 mm interval, and the transversal line was 40 mm from the left (Figure 4). For the boundary conditions, the right side was fixed, and the surfaces of the undulated plate were both free. Dynamic displacement was uniformly applied on the left side of the model, with a plate thickness of 2*h*. The transmitting signal can be expressed as x(t)=0.5Psin(2πft)(1−cos(2πft/N)), where f is the central frequency, N is the cycle of the tone burst, and P is the amplitude of the tone burst. In this simulation, *f* = 500 kHz, *N* = 10, and *P* = 0.0001 mm.

### 3.2. Results and Discussion

#### 3.2.1. Symmetrical Property

The displacement map of the Lamb waves propagating in the undulated plate based on the simulation is shown in Figure 5. It is evident that the desired S0 mode of the Lamb wave is effectively excited when the displacement loading is applied on the left.

All signals from the 11 detection points uniformly distributed at the intersecting line are received and analyzed to further verify the symmetrical property of the propagating Lamb waves. Figure 6 presents the received fundamental wave, the nonlinear wave, and the ZFC in the *x*_1_- and *x*_3_-directions, respectively. The displacements in the *x*_1_-direction along the thickness are symmetrical, whereas the displacements in the *x*_3_-direction is antisymmetric along the thickness. Because of the weak quadratic nonlinearity, the nonlinear waves excited by the transmitting signals with opposite phases are the same. Consequently, nonlinear waves can be obtained by removing the fundamental wave after summing the collected signals excited by two transmitting sources with opposite phases [62]. For the ZFC, it was obtained after the nonlinear waves were filtered with a low-pass filter. Figure 6 shows that all the received fundamental waves, nonlinear waves, and ZFCs are symmetrical.

#### 3.2.2. Material Early-Stage Damage Evaluation

Figure 7 shows the received signals at *x*_1_ = 40 mm and *x*_3_ = 0 mm in the frequency domain. Figure 6a presents the signal in the time domain. The magnitude of the ZFC (7.48 × 10^−7^) is more pronounced than that of the second harmonics (8.29 × 10^−8^). As discussed in Section 2, the magnitude of the nonlinear wave is utilized as the measured acoustic nonlinear parameter, which can indicate the material damage in the early stage.

As shown in Figure 8, the ZFC magnitude is plotted as a function of the propagating distance for different TOE. The ZFC magnitude monotonically increases with the propagating distance, which is consistent with the theoretical result (Figure 3). Considering that the varying tendency of the ZFC is the most crucial in this study, an arbitrary unit (a.u.) was employed for the magnitude. Degradations, including dislocations, persistent slip band, and microcracks, cause nonlinearities of solids and lead to material damage at an early stage. The degradation can be modeled with a higher magnitude of the TOE. Considering this, three models were established in this study, with the corresponding TOE being one, five, and ten times larger than those listed in Table 1, respectively. As shown in Figure 8, the magnitude of the curve slope increases with the magnitude of the TOE. Consequently, evaluating the early-stage damage of a symmetrical undulated plate based on a powerful ZFC is simple, with small errors, high reliability, high efficiency, and a long detection distance.

For comparison, a second harmonic–based evaluation methodology is also considered. Based on the theory discussed in Section 2 and Appendix A, the frequency of fundamental waves is set as 500 kHz for the damage inspection of the undulated plate. We set the spatial period distance ($L_{n}$) of the second harmonics equal to the spatial length (Λ = 40 mm) of the simulation undulated plate. Additionally, the frequency of the fundamental wave can be obtained as approximately 500 kHz by solving Equation (A1) in Appendix A. The frequency of the fundamental wave, as investigated in [24], can also be determined to be approximately 500 kHz, considering that $L_{n}$ = 40 mm. The generated and received signals for the three simulation models with 1TOE, 5TOE, and 10TOE are used to directly obtain the second harmonics. Magnitude of the second harmonic, as shown in Figure 9a, is presented as a function of the wave propagation distance. Consistent with the description in Appendix A and SQPM, the maximum cumulative wave propagation distance increases from approximately 20 mm to 100 mm.

Subsequently, the inspection distance is set to be approximately 100 mm. A linear fit of the received second harmonic is shown in Figure 9b, with a 5 mm distance between the adjacent detection points. Based on this figure, we can observe a high dispersion of data points. To improve the detection accuracy, as mentioned in Section 2, the distance between two adjacent detection points is set to 20 mm. As shown in Figure 9c, the discreteness of data points reduces to some extent, improving the detection accuracy. Evidently, evaluating the early-stage damage of a symmetrical undulated plate based on the second harmonics is feasible. However, this second harmonic–based inspection methodology is complicated, with large errors, low reliability, low efficiency, and a short detection distance. Thus, the use of a ZFC-based inspection methodology is recommended to evaluate the damage of an undulated plate in its early stage.

As shown in Figure 10, the magnitudes of the ZFC are presented as a function of wave propagating distances for different frequencies. The frequency ranges from 480 to 520 kHz with an interval of 5 kHz in Figure 10a, from 498 to 502 kHz with an interval of 1 kHz in Figure 10b, and from 500 kHz to 1 MHz with an interval of 100 kHz in Figure 10c. The magnitude of the ZFC changes with a similar tendency for different excitation frequencies, increasing with the frequency of the fundamental waves. Noises appear in Figure 10c when the frequency of the fundamental waves reaches a relatively high value, possibly reducing the detection precision. With an increasing frequency for the fundamental waves, a forbidden band, significant dispersion, and reflection will occur, which is not helpful in the evaluation of the undulated plate damage in its early stage. Therefore, the frequency of the fundamental waves is recommended to be lower than 700 kHz.

## 4. Discussions and Conclusions

The newly generated ZFCs and previously produced components at different times and places interfere with each other when they propagate through the same detection point of the undulated plate. Enhanced interference is maintained as long as the energy of the fundamental wave propagates through the plate regardless of the plate shape, with an increasing ZFC magnitude with the propagating distance of the fundamental waves. The study results show that the ZFC-based inspection methodology can be effectively utilized to evaluate the early-stage damage of a nonuniform plate.

As to the second harmonics, the newly generated and preceding ones interfere with each other at different times and places when they propagate through the same detection point. Enhanced interference occurs when the phases vary in the region of [0, *π*). By contrast, interference attenuation occurs when the phases vary in the region of [*π*, 2*π*). Consequently, SQPM was proposed to understand the complicated second-harmonic generation and propagation mechanisms, based on which a second harmonic–based inspection methodology could be established. Notably, since we did not carry out high-frequency validation of the SQPM technique, the presented conclusions are exclusively within the low-frequency domain, not exceeding 1 MHz. In practical applications, if the chosen frequency coincides with the band gap of the undulated plate, we advise against using second harmonics for damage detection. Conversely, zero frequency does not encounter this issue, which also underscores the advantage of zero-frequency detection for identifying damages in undulated plates.

This study, focusing on the undulated plate as an example, has theoretically and numerically demonstrated the robust ZFC structural immunity (ZFCSI). ZFCSI makes ZFC an ideal candidate for damage detection in complex geometries due to its accumulation with the propagation of fundamental wave energy, regardless of the structural shape or complexity. The lower carrier frequency of the ZFC enhances its penetrating power, allowing it to traverse through various complex geometries such as undulated plates, linearly varying thickness plates [68], and other plates with variable thickness. This inherent characteristic suggests that the ZFC could be a powerful tool for detecting damage in these complex structures.

In the future, to further validate the ZFCSI, we plan to undertake dedicated experimental studies. Given the challenges in densely and extensively extracting signals from complex structures, the implementation of noncontact laser scanning vibrometers for signal extraction is anticipated in these studies. Due to the presence of noise and interference in practical applications, it will be necessary to conduct numerous repeated measurements and take the average to stabilize the signal and reduce noise. Additionally, background signals will need to be measured and subtracted from the measured signals to eliminate noise and interference.

Utilizing the theoretical framework established in this paper, practical applications can be guided by the principles of ZFC accumulation. Taking the undulated plate as an example, the selection of signal extraction points need not be overly dense. It is sufficient to ensure that the distance between signal extraction points is equal, as the measured ZFC signals are theoretically distributed along a straight line with a positive slope. This approach not only simplifies the experimental setup but also enhances the reliability of the damage detection process in practice.

In the context of our study, which aims to detect the average density of early-stage damage along the straight line between the receiver and the transducer, the emphasis is primarily on the intensity of nonlinear signals in the frequency domain. This approach allows for a more straightforward [62] assessment of the damage severity without the complications associated with time-varying effects. As depicted in Figure 6, we have included an example of a time domain signal to illustrate the nature of the signals we are analyzing. However, the time domain features were not considered in the theoretical modeling. Time domain information can be crucial for distinguishing between different types of damage. This is an area of interest that we are keen to explore in future research, where we aim to delve deeper into the characteristics of nonlinear signals in the time domain.

In conclusion, the propagation of nonlinear Lamb waves, including the ZFC and second harmonics, in a symmetrical undulated plate was theoretically and numerically investigated. Moreover, an application in monitoring the material damage in the undulated plate in its early stage was demonstrated. This shows that the second harmonic inspection method is comparatively complicated and limited in engineering applications. Compared to the second harmonic inspection technique, the ZFC-based inspection methodology is easier to implement because the ZFC exhibits characteristics independent of the geometry, with small errors, high reliability, high efficiency, broadened discretion in frequency selection, and a long detection distance. Therefore, the promising ZFC-based inspection technique can be used to evaluate the material damage of a nonuniform plate in its early stage.

## Figures and Tables

**Figure 1 sensors-24-04878-f001:**
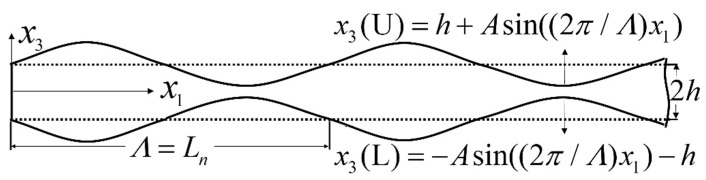
Geometry of the periodically undulated isotropic plate.

**Figure 2 sensors-24-04878-f002:**
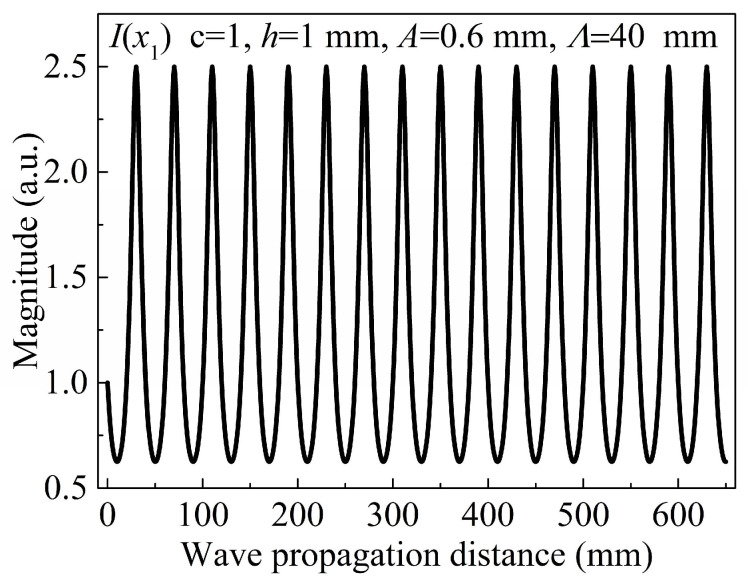
Plotted as a function of the propagating distance.

**Figure 3 sensors-24-04878-f003:**
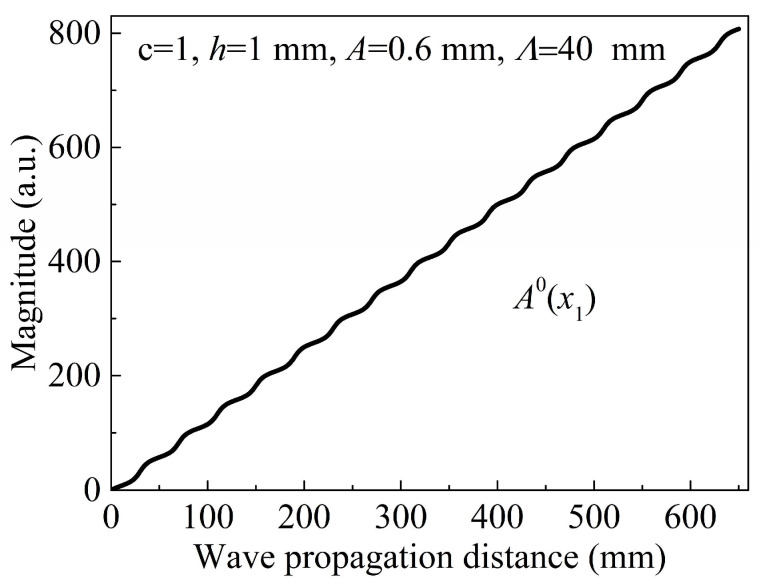
Plotted as a function of the wave propagation distance.

**Figure 4 sensors-24-04878-f004:**
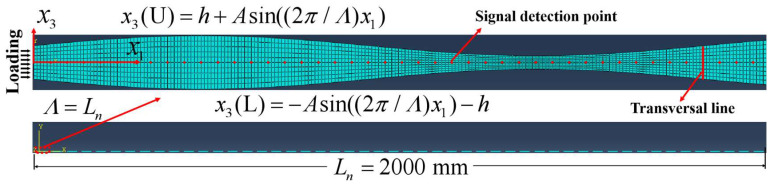
(Color online) Schematic of the simulation model of a 2D periodically undulated plate.

**Figure 5 sensors-24-04878-f005:**
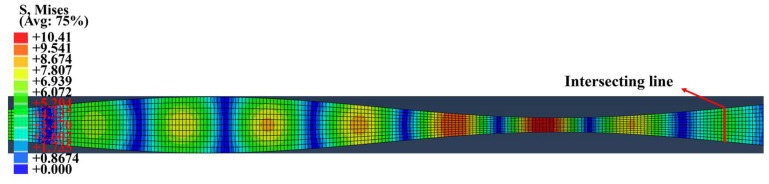
(Color online) Displacement map of the simulation model of the 2D periodically undulated plate.

**Figure 6 sensors-24-04878-f006:**
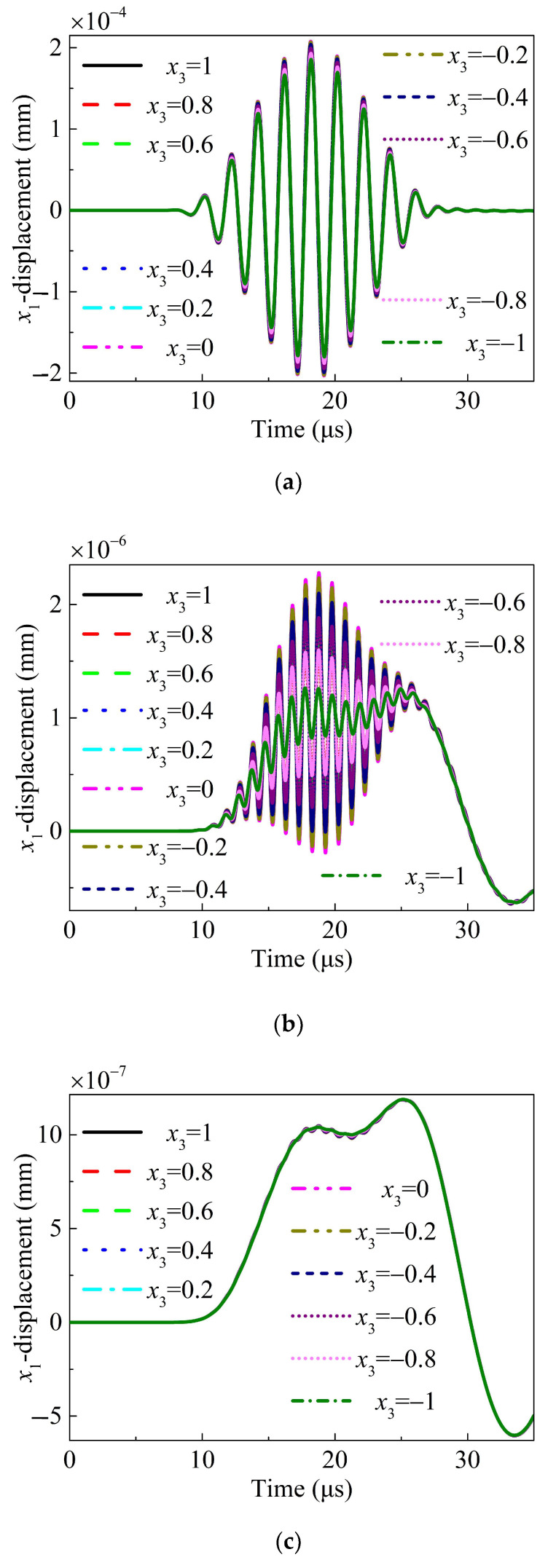
(Color online) Displacement at points on the same cross section along the transversal line (*x*_1_ = 40 mm). (**a**) Wave received in the *x*_1_-direction; (**b**) the obtained nonlinear wave of the received wave in the *x*_1_-direction by removing the fundamental wave; (**c**) the obtained ZFC of the nonlinear wave in the *x*_1_-direction by removing the second harmonic; (**d**) wave received in the *x*_3_-direction; (**e**) the obtained nonlinear wave of the received wave in the *x*_3_-direction by removing the fundamental wave; (**f**) and the obtained ZFC of the nonlinear wave in the *x*_3_-direction by removing the second harmonic.

**Figure 7 sensors-24-04878-f007:**
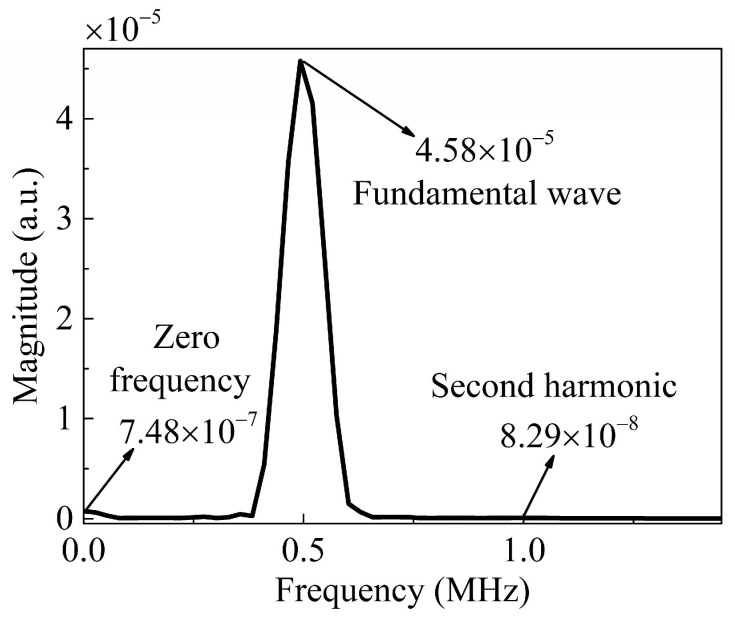
Signals received at *x*_1_ = 40 mm and *x*_3_ = 0 mm in the frequency domain.

**Figure 8 sensors-24-04878-f008:**
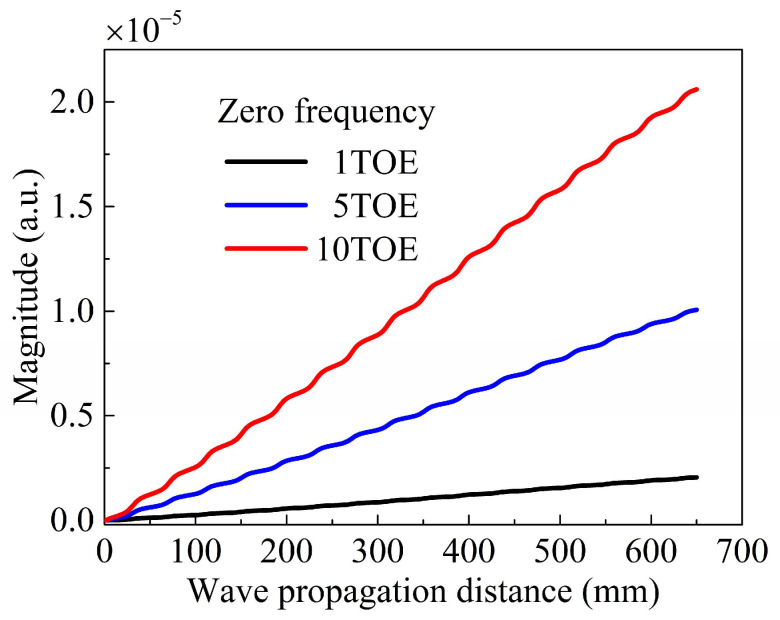
(Color online) Magnitude of the zero-frequency component (ZFC) as a function of wave propagating distances for different TOE.

**Figure 9 sensors-24-04878-f009:**
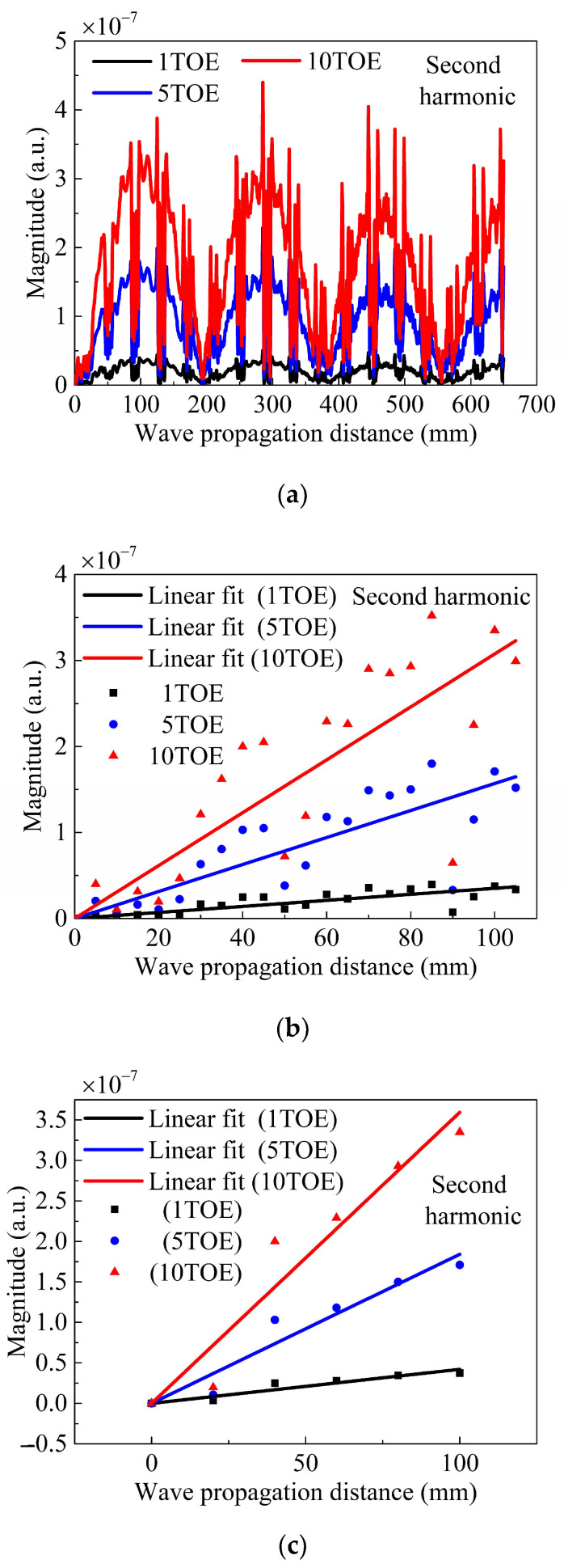
(Color online) Magnitude of the second harmonic as a function of wave propagation distance for different TOE. (**a**) The distance between two signal detection points is 1 mm (from 0 to 650 mm); (**b**) the distance between two signal detection points is 5 mm (from 0 to 105 mm); (**c**) the distance between two signal detection points is 20 mm (from 0 to 100 mm).

**Figure 10 sensors-24-04878-f010:**
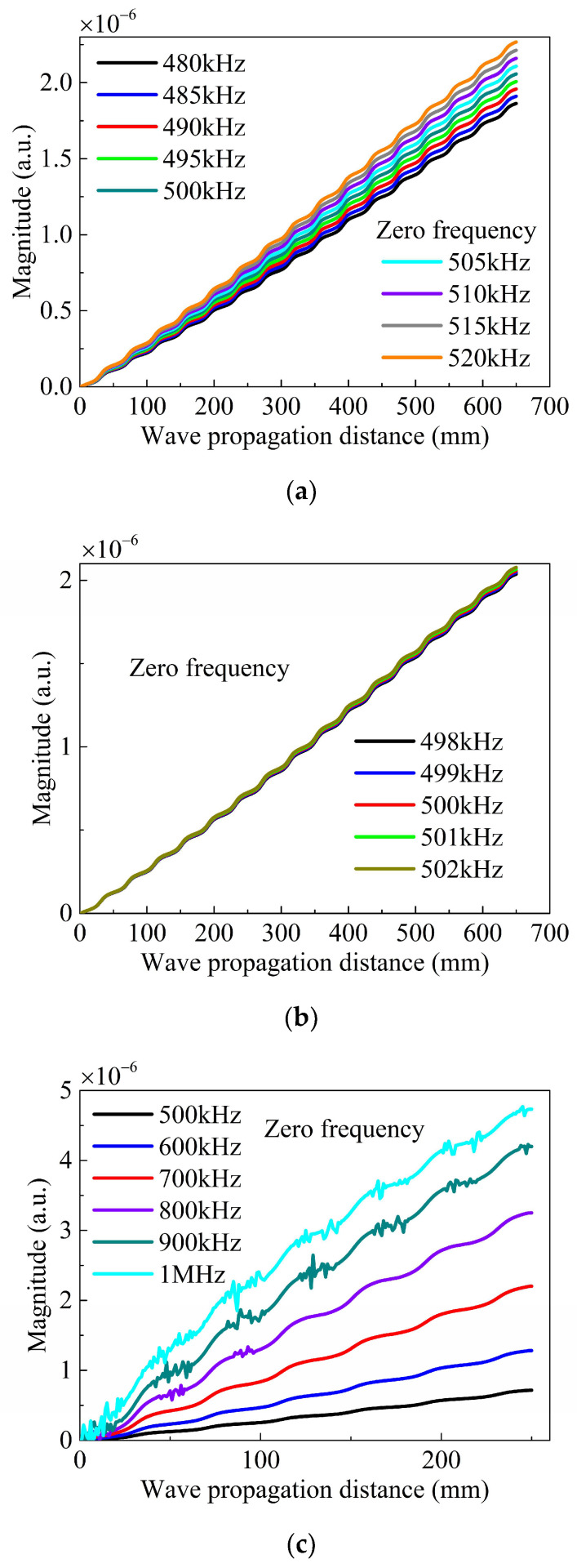
(Color online) Magnitude of the zero-frequency component as a function of wave propagating distances for different frequencies. (**a**) 480–520 kHz at intervals of 5 kHz; (**b**) 498–502 kHz at intervals of 1 kHz; (**c**) 500–1000 kHz at intervals of 100 kHz.

**Table 1 sensors-24-04878-t001:** Physical parameters of the Al-7075-T651 plate.

*ρ* (kg/m^3^)	A (GPa)	B (GPa)	C (GPa)	*λ* (GPa)	*µ* (GPa)	*E* (GPa)	*ν*
2704	−416	−131	−150.5	70.3	26.96	68.9	0.33

## Data Availability

The data supporting the findings of this study are available from the corresponding author upon request.

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
