# Peer review of "Investigation of the Zero-Frequency Component of Nonlinear Lamb Waves in a Symmetrical Undulated Plate"

_sensors, 2024, doi:10.3390/s24154878_

Round 1

Reviewer 1 Report

Comments and Suggestions for Authors

I would like to express my appreciation to the authors for submitting their manuscript on the investigation of the zero-frequency component of nonlinear Lamb waves in a symmetrical undulated plate. The topic of this research is highly relevant to the field of NDT, particularly in the context of early-stage damage assessment. Below are my specific comments and suggestions regarding this manuscript:

Originality and Significance: The study provides an in-depth theoretical and numerical investigation of the ZFC of nonlinear Lamb waves, offering new insights into early damage detection, which is crucial for structural safety in engineering applications.

Methodological Rigor: The combination of theoretical analysis and numerical simulation is a robust approach that effectively validates theoretical assumptions and tests them in practical scenarios.

Reliability of Results: The paper presents a comparative study between the ZFC and the second harmonic component (SHC) in terms of their efficiency and capability in damage detection, demonstrating the advantages of the ZFC method for symmetrical undulated plate inspection.

Clarity of Figures and Data Presentation: The figures and data in the paper are clear and presented in an intuitive manner, aiding the reader in understanding the research outcomes.

Suggestions for Improvement:

Experimental Validation: If possible, including some experimental data to complement the simulation results would strengthen the paper's conclusions.

Expansion of Discussion: Although the authors have discussed the advantages of the ZFC method, further discussion on how to overcome practical challenges such as environmental noise and operational complexity would be valuable.

Specificity of Conclusions: The conclusions could more specifically outline the application prospects of the ZFC method in real-world engineering problems and suggest directions for future research.

Overall, I believe this manuscript is valuable to the readers of the "Sensors" journal as it provides a novel perspective on understanding and applying the role of nonlinear Lamb waves in SHM. Based on the above evaluation, I recommend that the paper be considered for publication after appropriate revisions.

Comments on the Quality of English Language

I did not found any inappropriate use of English.

Author Response

Please refer to the attached Response letter. Thanks!

Reviewer 2 Report

Comments and Suggestions for Authors

In this manuscript, the authors illustrated the utilization of zero-frequency component of lamb waves in an undulated plate. Theoretical basis and simulations were presented for the illustration.

1 As zero-frequency Component was focused, drifting in the signals in real applications would influence the results.

2 Note Lamb wave was focused, the time domain features were not considered in the theoretical modeling.

3 Why the magnitude of waves increases in Figure 3? What is the physical meaning?

4 Please explain the structural quasi-phase-matching (SQPM) technique in detail.

5 What is TOE and its physical meaning?

6 The sizes of equations are much larger than the main content.

Comments on the Quality of English Language

Moderate editing of English language required

Author Response

(The authors gave the same response as above.)

Reviewer 3 Report

Comments and Suggestions for Authors

The manuscript describes the behaviour of the zero-frequency component of nonlinear 2 Lamb waves in symmetrical undulated plates. It is a very specific investigation, but it is well done and clearly presented. My opinion is to publish in the present form.

Author Response

(The authors gave the same response as above.)

Round 2

Reviewer 2 Report

Comments and Suggestions for Authors

The authors have addressed all the issues

Comments on the Quality of English Language

The authors have addressed all the issues